# Protein-responsive protein release of supramolecular/polymer hydrogel composite integrating enzyme activation systems

Hajime Shigemitsu [1,4,5], Ryou Kubota [1,5], Keisuke Nakamura [1], Tomonobu Matsuzaki[1], Saori Minami[2], Takuma Aoyama[2], Kenji Urayama [2] & Itaru Hamachi [1,3✉]

Non-enzymatic proteins including antibodies function as biomarkers and are used as biopharmaceuticals in several diseases. Protein-responsive soft materials capable of the controlled release of drugs and proteins have potential for use in next-generation diagnosis and therapies. Here, we describe a supramolecular/agarose hydrogel composite that can release a protein in response to a non-enzymatic protein. A non-enzymatic protein-responsive system is developed by hybridization of an enzyme-sensitive supramolecular hydrogel with a protein-triggered enzyme activation set. In situ imaging shows that the supramolecular/agarose hydrogel composite consists of orthogonal domains of supramolecular fibers and agarose, which play distinct roles in protein entrapment and mechanical stiffness, respectively. Integrating the enzyme activation set with the composite allows for controlled release of the embedded RNase in response to an antibody. Such composite hydrogels would be promising as a matrix embedded in a body, which can autonomously release biopharmaceuticals by sensing biomarker proteins.

[1] Department of Synthetic Chemistry and Biological Chemistry, Graduate School of Engineering, Kyoto University, Nishikyo-ku, Katsura, Kyoto 615-8510, Japan. [2] Department of Macromolecular Science and Engineering, Kyoto Institute of Technology, Matsugasaki, Kyoto 606-8585, Japan. [3] JST-ERATO, Hamachi Innovative Molecular Technology for Neuroscience, Kyoto University, Nishikyo-ku, Kyoto 615-8530, Japan. [4] Present address: Department of Applied Chemistry, Graduate School of Engineering, Osaka University, 2-1 Yamadaoka, Suita, Osaka 565-0871, Japan. [5] These authors contributed equally: Hajime Shigemitsu, Ryou Kubota. ✉email: ihamachi@sbchem.kyoto-u.ac.jp

Proteins are one of the pivotal biomolecules necessary for life. Proteins often function as biomarkers and can be used as biopharmaceuticals for many diseases[1]. Soft materials capable of controlled drug release in response to biomarker proteins are becoming increasingly important for next-generation diagnosis, drug delivery systems, and therapies[2,3]. Stimulus-sensitive hydrogels are highly promising scaffolds for the detection of biomarker proteins because of their biocompatibility and chemical programmability[4]. The finely tunable physicochemical properties of these hydrogels also enable the controlled release of embedded small-molecule or protein-based drugs after implantation or injection in vivo[5–12]. Although several hydrogels have been demonstrated to respond to reactive stimuli, including redox reagents, light, and enzymes[13–22], it is still a challenge to rationally design hydrogels that respond to non-enzymatic proteins. Such non-enzymatic protein-responsive hydrogels are expected to be useful for controlled release of biopharmaceuticals in response to disease-related biomarker proteins such as antibodies, secreted cytokines, and membrane receptors. Miyata et al. have pioneered the development of antigen-responsive polymer hydrogels based on competitive antibody–antigen recognition between the target and the corresponding antibody–antigen pair[23]. The recognition-based mechanism was recently extended to other recognition pairs, including protein–DNA aptamer and membrane receptor–growth factor pairs[24–26]. Despite being potentially useful, these typically suffer from low sensitivity because they rely on a 1:1 recognition mechanism and, in many cases, their volume change is small. Controlled release of embedded proteins by coupling the release with the recognition of a specific protein has not yet been demonstrated; however, the release of small molecules and nanoparticles from hydrogel matrices has been evaluated. The development of a hydrogel scaffold having a robust mechanism responsive to non-enzymatic proteins is highly desirable.

We herein describe the development of non-enzymatic protein-responsive soft materials by integrating an enzyme-sensitive supramolecular hydrogel with a protein-triggered enzyme activation system (Fig. 1a). To convert an input of a non-enzymatic protein into an enzyme activity, we design enzyme-activity triggers (EATs) consisting of an enzyme inhibitor and the ligand of a target protein that are linked with a short linker. Hybridization of the enzyme-sensitive supramolecular hydrogel, an enzyme, and the designer EAT shows a macroscopic gel–sol transition responsive to a target protein. Moreover, this system is mixed with agarose gel to produce a supramolecular/polymer hydrogel composite with the protein response ability. In situ confocal laser scanning microscopic (CLSM) imaging reveals that the composite hydrogel consists of orthogonal domains of supramolecular fibers and agarose, which play distinct roles in protein entrapment and mechanical stiffness, respectively. We also succeed in the controlled release of the embedded RNase in response to an antibody from the composite hydrogel.

## Results

### Design of a non-enzymatic protein-responsive hydrogel system
Peptide-based supramolecular hydrogels consisting of self-assembled nanofibers were designed that would undergo a macroscopic gel–sol transition by an enzyme trigger. Two diphenylalanine derivatives were synthesized, which had tethered acetoxybenzyl-oxycarbonyl or benzoate groups at the N-terminus as an essential part of hydrogel formation (APmoc-F(CF₃)F and Bz-FF, respectively) (Fig. 1b). A decrease in the hydrophobicity of these compounds, through the enzymatic cleavage of the N-terminal moiety (and spontaneous 1,6-elimination in the case of APmoc-F(CF₃)F), is expected to lead to the

destabilization of the supramolecular nanofibers and subsequent collapse of the hydrogel. For an enzyme activation triggered by a target protein, we prepared sets of EATs and a corresponding enzyme (Fig. 1c)[27]. Using such pairs can convert an input (a non-enzymatic target protein) into enzymatic activity (Fig. 1a); thereby, a gelator can be decomposed to induce a macroscopic gel–sol transition. The EAT consists of the ligand of a target protein and an enzyme inhibitor, which are connected by a short linker. We used bovine carbonic anhydrase II (bCAII) as the enzyme, which has activity that is tentatively inhibited by EATs. Upon addition of the target protein, the EAT would preferentially bind to the protein and release bCAII because of steric repulsion, resulting in recovery of the bCAII activity to facilitate the degradation of the hydrogelators.

### bCAII response of the peptide-type hydrogels
We initially examined the susceptibility of the two hydrogelators toward bCAII using a hydrogel array chip (Fig. 2a). We prepared APmoc-F(CF₃)F and Bz-FF hydrogel droplets on glass slides, and then added an aqueous solution of bCAII. After 6 h, the state (gel or sol) was evaluated by the water absorption test with a paper. An APmoc-F(CF₃)F hydrogel (0.35 wt%; critical gelation concentration (CGC) 0.20 wt%, Supplementary Fig. 2) successfully turned into the sol state after addition of bCAII (Fig. 2b, lane 2; Supplementary Fig. 3). Upon addition of a solution lacking bCAII, or containing bCAII premixed with an inhibitor (ethoxzolamide, EZA), the gel–sol transition did not occur (Fig. 2b, lanes 1 and 3, respectively). These results indicated that APmoc-F(CF₃)F is a bCAII-responsive hydrogelator. In contrast, Bz-FF was not hydrolyzed at the benzoate ester moiety, because it is an unfavorable substrate for bCAII, and thus the hydrogel (2.0 wt%; CGC 1.5 wt%, Supplementary Fig. 4) remained in the gel state on addition of bCAII (Fig. 2b, lane 5; Supplementary Fig. 5).

### Avidin-response by integrating an enzyme activation system
We selected avidin as a target protein for use in a proof-of-principle trial and designed the corresponding EAT(avidin). According to the method described by Tan[27], we linked biotin, a strong ligand for avidin ($K_d$: 1 fM), and benzenesulfonamide, a bCAII inhibitor ($K_d$: ca. 1 μM)[28,29], via a short ethylene linker (Fig. 1c). Because the linker length (ca. 5 Å) was much shorter than the distance from the catalytic $Zn^{2+}$ center to the surface of bCAII (ca. 19 Å), it is reasonable to expect that EAT(avidin) will not be able to inhibit bCAII after it is bound to avidin (Supplementary Fig. 6). The APmoc-F(CF₃)F hydrogel containing bCAII and EAT(avidin) was prepared by adding a mixture of bCAII and EAT(avidin) (at a 1:2 ratio) to the APmoc-F(CF₃)F before gelation. We found the hydrogel changed to the sol state 6 h after the addition of avidin (1.0 equiv. versus EAT(avidin)), (Fig. 2c, lane 2). In contrast, the hydrogel did not show a gel–sol transition on treatment with either buffer solution or by the addition of avidin together with biotin (Fig. 2c, lanes 1 and 3, respectively; hereafter these two conditions are referred to as the control conditions). Furthermore, the APmoc-F(CF₃)F hydrogel without bCAII/EAT(avidin) retained the hydrogel state upon addition of avidin, suggesting that the bCAII–EAT(avidin) pair is crucial for an avidin-induced gel–sol transition (Fig. 2c, lane 4). The chemical degradation of APmoc-F(CF₃)F triggered by avidin was evaluated by high-performance liquid chromatography (HPLC) and CLSM[30]. HPLC analysis indicated that 80% of the APmoc-F(CF₃)F was decomposed after 6 h, resulting in a residual gelator concentration (0.07 wt%) that was lower than the CGC of APmoc-F(CF₃)F (0.20 wt%) (Fig. 2d, Supplementary Fig. 7). Conversely, the decomposition of APmoc-F(CF₃)F was negligible under the control conditions. CLSM imaging of the APmoc-F

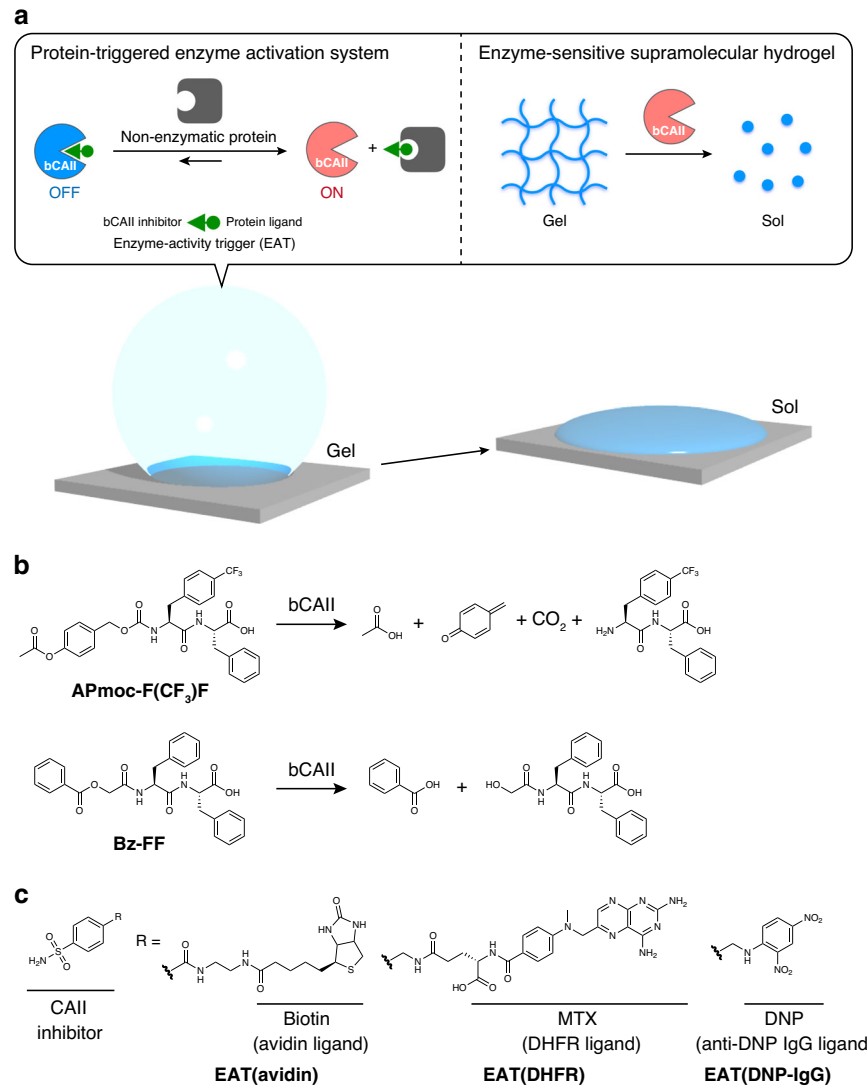

**Fig. 1 Design of a non-enzymatic protein-responsive supramolecular hydrogel. a** Schematic illustration of a non-enzymatic protein-responsive soft material consisting of an enzyme (bCAII)-sensitive supramolecular hydrogel and a protein-triggered bCAII activation system that converts an input signal (a non-enzymatic protein) into the enzymatic activity via enzyme-activity trigger (EAT). bCAII: bovine carbonic anhydrase II. **b** Chemical structures and plausible reaction schemes of **APmoc-F(CF₃)F** and **Bz-FF** with bCAII. **APmoc-F(CF₃)F** is a bCAII-responsive hydrogelator. **APmoc-F(CF₃)F** showed the lowest critical gelation concentration (0.20 wt%) than other APmoc gelators we tested (**APmoc-FF**, **-F(F)F**, **-FF(F)**, **-F(F)F(F)**, **-FF(CF₃)**, and **-F(CF₃)F (CF₃)**) (Supplementary Fig. 1). **c** Chemical structures of **EAT(avidin)**, **EAT(DHFR)**, and **EAT(DNP-IgG)**. DHFR: dihydrofolate reductase, DNP: dinitrophenyl, MTX: methotrexate.

(CF₃)F hydrogel with bCAII/**EAT(avidin)** after staining with an appropriate fluorescent probe (**TMR-Gua**, Supplementary Fig. 8) allowed visualization of the well-elongated nanofibers of **APmoc-F(CF₃)F**. These nanofibers completely disappeared 6 h after addition of avidin, while the nanofibers remained intact under the control conditions (Fig. 2e). Taken together, these results indicate that a supramolecular hydrogel responsive to a non-enzymatic protein (avidin) was successfully constructed through hybridization of bCAII-responsive **APmoc-F(CF₃)F** and the protein-triggered bCAII activation set.

We also evaluated the threshold amount of avidin required for the gel–sol transition of the hydrogel droplets (Fig. 2f). The use of a gel array chip showed that a minimum of 200 pmol of avidin was required to induce the gel–sol transition. HPLC analysis demonstrated that 48% of **APmoc-F(CF₃)F** (52.6 μmol) was decomposed at this threshold level (Supplementary Fig. 9); this suggested that the avidin-triggered bCAII activation system could amplify the input avidin signal by ca. 260-fold. The threshold-

type macroscopic gel–sol response was further supported by rheological and CLSM analyses (Supplementary Figs. 10, 11). When we synthesized an EAT with a longer oligoethylene glycol linker (ca. 60 Å) (**EAT(avidin, long)**) (Supplementary Fig. 12), both the hydrogel array chip and HPLC analyses showed that the bCAII activity was not effectively recovered upon addition of avidin as shown in Supplementary Fig. 13. This non-recovery can be presumably ascribed to a lack of steric repulsion between bCAII and avidin, to form a stable ternary complex bCAII/EAT/avidin.

**Gel–sol response towards other target proteins.** In our strategy, a simple change in the protein ligands of the EATs allowed proteins other than avidin to be targeted. For instance, the proteins dihydrofolate reductase (DHFR) or anti-dinitrophenyl (DNP) IgG could be targeted (Supplementary Figs. 14, 15) by the synthesis of two different EATs containing the corresponding

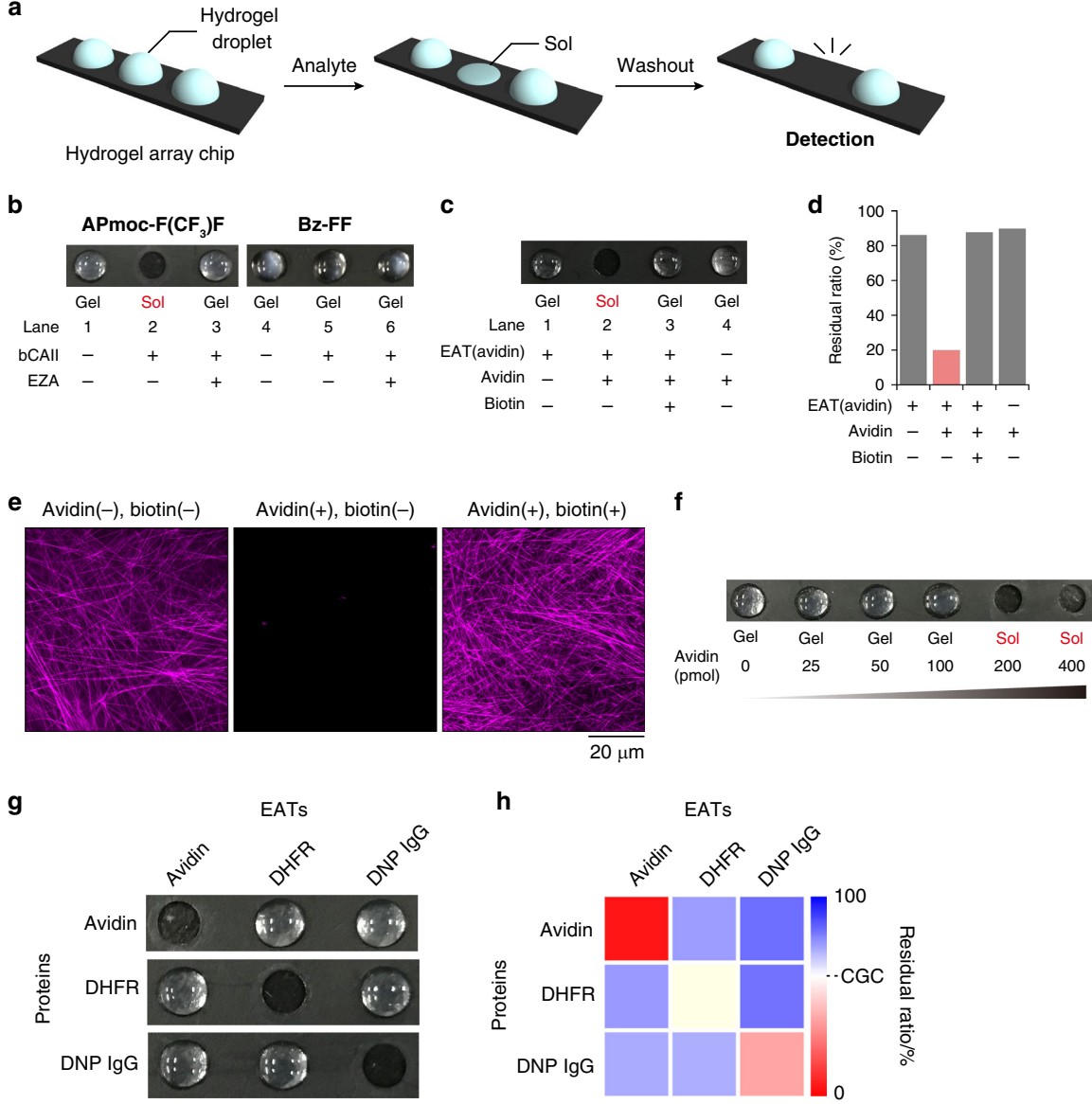

**Fig. 2 bCAII and protein responses of supramolecular hydrogels. a** Schematic illustration of naked eye detection of a gel–sol transition triggered by bCAII and non-enzymatic proteins on a hydrogel array chip. **b** bCAII response of (left) **APmoc-F(CF₃)F** and (right) **Bz-FF**. EZA: ethoxzolamide. Condition for bCAII response: [**APmoc-F(CF₃)F**] = 0.35 wt% (6.1 mM), [**Bz-FF**] = 2.0 wt% (42 mM), [bCAII] = 10 μM, [EZA] = 100 μM, 100 mM HEPES (pH 8.0), 25 °C, 6 h, $V_{gel}$:$V_{stimulus}$ = 10:1. **c** Avidin response of **APmoc-F(CF₃)F** hydrogels containing bCAII and **EAT(avidin)**. **d** Residual ratios of **APmoc-F(CF₃)F** after treatment of avidin determined by HPLC. The background hydrolysis of the acetyl group occurred probably because the enzymatic activity of bCAII was not completely inhibited (Supplementary Fig. 7). **e** High-resolution Airyscan CLSM images of **APmoc-F(CF₃)F** hydrogels containing bCAII and **EAT(avidin)** with a fluorescent probe, **TMR-Gua**, after addition of (left) buffer, (middle) avidin, and (right) the mixture of avidin and biotin. **f** Determination of the detection threshold. **g** The non-enzymatic protein response in various combinations of target proteins and EATs. **APmoc-F(CF₃)F** hydrogels changed to the sol state only under the appropriate pairs. **h** Heat map of the residual ratios of **APmoc-F(CF₃)F** after addition of analytes determined by HPLC analysis. CGC: Critical gelation concentration. Condition for non-enzymatic protein response: [**APmoc-F(CF₃)F**] = 0.35 wt% (6.1 mM), [bCAII] = 10 μM, [**EAT (avidin)**] = 20 μM, [**EAT(DHFR)**] = 45 μM, [**EAT(DNP-IgG)**] = 15 μM, [avidin] = 20 μM (for **c**, **d**, **e**, **g**, and **h**) and 0, 1.25, 2.5, 5.0, 10, 20 μM (for **f**), [biotin] = 120 μM, [DHFR] = 45 μM (for **g**, **h**), [DNP IgG] = 15 μM (for **g**, **h**), [**TMR-Gua**] = 10 μM (for **e**), 100 mM HEPES, pH 8.0, 25 °C, 6 (for **c**, **d**, and **e**), 12 (for **f**), and 18 h (for **g** and **h**), $V_{gel}$:$V_{stimulus}$ = 10:1.

ligands methotrexate (MTX) (**EAT(DHFR)**) or DNP (**EAT (DNP-IgG)**), respectively (Fig. 1c). Both the gel droplet experiments and HPLC analyses clearly showed that DHFR and anti-DNP IgG-responsive hydrogels could be constructed by incorporation of the corresponding EATs (Supplementary Figs. 14, 15, 16, 17). To demonstrate the orthogonality of the protein response, we further investigated the responses with different combinations of EATs and target proteins (bCAII/**EAT(avidin or DHFR or DNP-IgG)⊂APmoc-F(CF₃)F**). As shown in Fig. 2g, the hydrogels

successfully exhibited a gel–sol transition only under the combination of a target protein and its specific EAT. HPLC analyses also showed that **APmoc-F(CF₃)F** was degraded only in the case of the appropriate pairs (Fig. 2h, Supplementary Fig. 18). These results clearly revealed that this method is a robust strategy for synthesizing non-enzymatic protein-responsive hydrogels.

**Supramolecular–polymer composite hydrogel.** Having a non-enzymatic protein-responsive supramolecular hydrogel system in

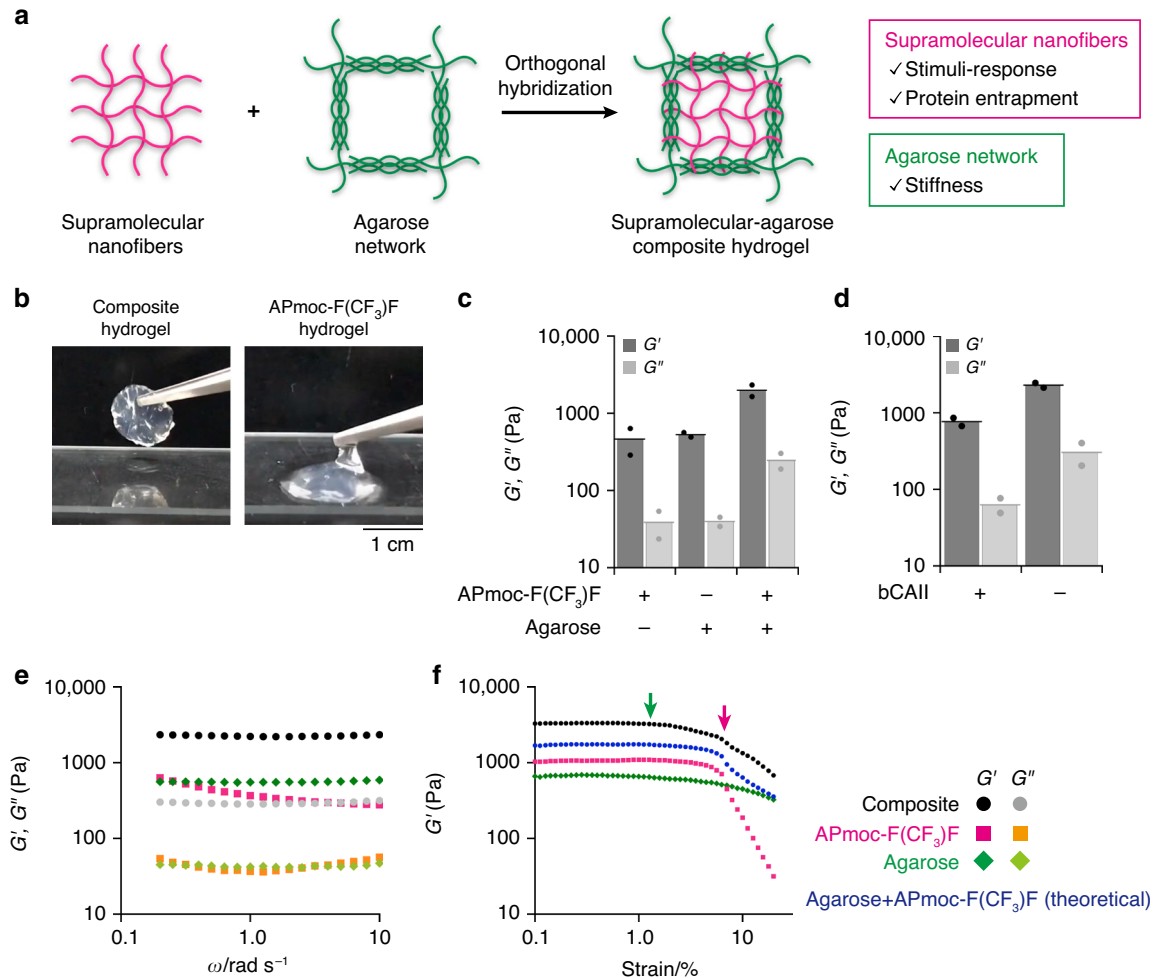

**Fig. 3 Rheological analysis of a supramolecular–agarose composite hydrogel. a** Schematic illustration of orthogonal hybridization of supramolecular **APmoc-F(CF₃)F** hydrogel and agarose gel. **b** Photographs of (left) the composite hydrogel and (right) the **APmoc-F(CF₃)F** hydrogel. **c** Rheological properties of (left) **APmoc-F(CF₃)F**, (middle) agarose, and (right) composite hydrogels. The data represent the mean ($n = 2$). **d** Rheological properties of the composite hydrogel after addition of (left) bCAII and (right) buffer lacking bCAII. Sweep rate: 0.2 rad/s, strain: 1%. **e** Frequency sweep rheological properties of the (magenta) **APmoc-F(CF₃)F**, (green) agarose, and (black) composite hydrogels. Strain amplitude: 1%. **f** $G'$ values of (magenta) **APmoc-F(CF₃)F**, (green) agarose, (black) composite gels, and (blue) the sum of **APmoc-F(CF₃)F** and agarose. Green and magenta arrows indicate the limit points of the linear viscoelastic region of agarose and **APmoc-F(CF₃)F**. The $G'$ value of the composite hydrogel changed in two steps around the green and magenta arrows. Frequency: 10 rad/s. $G'$: storage shear modulus, $G'$: loss shear modulus. Condition: [**APmoc-F(CF₃)F**] = 0.6 wt%, [agarose] = 0.5 wt%, [bCAII] = 30 μM, 8 h, rt.

hand, we next attempted to design an intelligent soft material capable of releasing a protein in response to another protein. However, the mechanical toughness of the **APmoc-F(CF₃)F** hydrogel was found not to be sufficient for such a matrix. Composite materials for supramolecular and polymer hydrogels have recently been proposed to overcome the mechanical weakness of supramolecular hydrogels (Fig. 3a)[31–64]. It was expected that the supramolecular fibers and the polymer network would play distinct roles, that is protein entrapment and stimulus responsiveness will be a result of the supramolecular fibers and mechanical stiffness will be imparted by the polymer network. However, the potential utility of such composite hydrogels has not yet been demonstrated well because of insufficient structural and functional analyses. Agarose was employed as a polymer gel because the gel can be prepared by a protocol similar to that for the **APmoc-F(CF₃)F** hydrogel (Supplementary Fig. 19). The opaque composite hydrogel was obtained by mixing hot agarose solution and **APmoc-F(CF₃)F** powder in aqueous HEPES buffer (pH 8.0), and then heating until the powder dissolved, followed

by cooling to rt. The resultant composite hydrogel showed enough mechanical stiffness to be able to be picked up with tweezers (Fig. 3b, left; Supplementary movie 1). In contrast, the **APmoc-F(CF₃)F** hydrogel could not retain a shape without a mold (Fig. 3b, right). Rheological analyses quantitatively demonstrated that the storage modulus ($G'$) of the composite hydrogel was synergistically enhanced to be 2000 Pa, which was higher than that of the agarose hydrogel (530 Pa) or supramolecular **APmoc-F(CF₃)F** hydrogel (400 Pa) (Figs. 3c, 3e, Supplementary Fig. 20).

CLSM imaging of the composite hydrogel provided interesting structural insights. We used fluorescein-modified agarose (FL-agarose) and **APmoc-F(CF₃)F** stained with **TMR-Gua** (FL-agarose: 0.5 wt%, **APmoc-F(CF₃)F**: 0.6 wt%). As shown in Fig. 4a, CLSM successfully visualized the fibrous networks of **APmoc-F (CF₃)F** even in the composite hydrogel (Fig. 4a, left). In contrast, the agarose network presented as a sea-island pattern (Fig. 4a, middle), which may correspond to the aggregated double-helix model proposed for agarose networks[65]. Surprisingly, the overlay

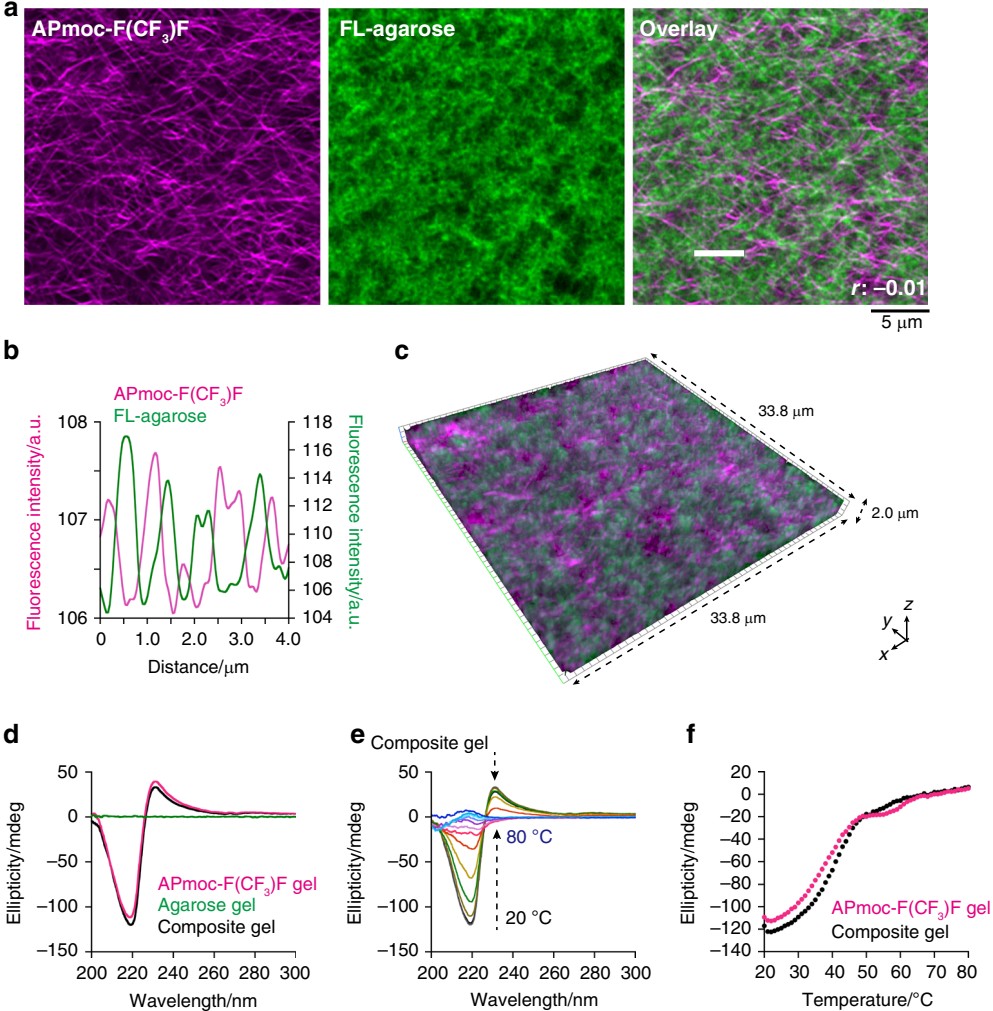

**Fig. 4 Structural characterization of the supramolecular–agarose composite hydrogel. a**, **c** High-resolution Airyscan CLSM images of the composite hydrogel composed of FL-agarose and **APmoc-F(CF₃)F** stained with **TMR-Gua**. The staining selectivity was confirmed as shown in Supplementary Fig. 21. Left: TMR channel, middle: fluorescein channel, right: the overlay image. Condition: [**APmoc-F(CF₃)F**] = 0.6 wt%, [FL-agarose] = 0.5 wt%, [**TMR-Gua**] = 10 μM, 100 mM HEPES, pH 8.0. *r*: Pearson's correlation coefficient. **b** Line plot analysis of fluorescent intensity along a white line shown in Fig. 4a. Magenta: **APmoc-F(CF₃)F**, green: FL-agarose. **d** CD spectra of the (magenta) **APmoc-F(CF₃)F**, (green) agarose, and (black) composite hydrogels. Temperature: 25 °C. **e** Temperature-dependence CD spectra of the composite hydrogel. Temperature interval: 5 °C. **f** Plots of the CD intensity at 220 nm of the (magenta) **APmoc-F(CF₃)F** and (black) composite hydrogels. Condition: [**APmoc-F(CF₃)F**] = 0.6 wt%, [agarose] = 0.5 wt%, 100 mM HEPES, pH 8.0, optical length: 0.05 mm.

image showed the two components were segregated rather than coassembled, that is the **APmoc-F(CF₃)F** nanofibers were mainly located at the darker regions of the void spaces of the agarose network (Fig. 4a, right). This observation was supported by the Pearson's correlation coefficient[66] (–0.01) and the line plot analysis, which showed that the tops of the peaks did not overlap with each other (Fig. 4b). The z-stack CLSM image demonstrated that the **APmoc-F(CF₃)F** nanofibers and the agarose network were well entangled but did not overlap in the three dimensional space (Fig. 4c, Supplementary Fig. 22, Supplementary movie 2). Also, the individual morphologies of **APmoc-F(CF₃)F** and agarose in the composite hydrogel were quite similar to those in the single-component **APmoc-F(CF₃)F** and agarose hydrogels, implying that interactions between **APmoc-F(CF₃)F** and agarose were minimal (Supplementary Fig. 23). The orthogonality of **APmoc-F(CF₃)F** and agarose was further supported by CD spectroscopy and rheological analysis. CD spectroscopy indicated that the single-component **APmoc-F(CF₃)F** and the composite hydrogels showed an almost identical negative Cotton peak at

218 nm and quite similar temperature-dependent spectral changes (Figs. 4d–f, Supplementary Fig. 24). These results indicated that the packing structure and self-assembly property of **APmoc-F(CF₃)F** in the composite hydrogel is the same as in the single-component **APmoc-F(CF₃)F** gel. Strain-sweep rheological analyses demonstrated that the storage modulus of the single-component **APmoc-F(CF₃)F** gel sharply decreased at 4.0% strain, while that of the agarose gel gradually decreased from 1.0% strain (Fig. 3f, Supplementary Fig. 20a, 20b). The change of the storage modulus of the composite hydrogel was almost similar to the sum of **APmoc-F(CF₃)F** and agarose gels, that is, gradually decreased from 1.1% strain and the decrease rate clearly changed at 4.5% strain (Fig. 3f, Supplementary Fig. 20c). These data suggested that strain responses of **APmoc-F(CF₃)F** nanofibers and agarose networks near the linear viscoelastic region orthogonally retained in the composite hydrogels. Taken together, these results indicated that **APmoc-F(CF₃)F** and agarose form orthogonal networks with negligible interference in the composite hydrogel.

We also conducted scanning electron microscopic (SEM) analysis of the single-component **APmoc-F(CF₃)F** or agarose, and the composite hydrogels. SEM images of **APmoc-F(CF₃)F** and agarose gels showed similar well-entangled nanofiber structures (diameter: ca. 10–20 nm)[50], so that it is difficult to distinguish **APmoc-F(CF₃)F** and agarose fibers (Supplementary Fig. 25a, b). On the other hand, the composite hydrogel exhibited the completely distinct morphologies from the single-component hydrogels, that is two-dimensional sheets and thicker bundled structure (diameter: ca. 200 nm), which was probably induced as an artifact during a drying process of the SEM sample preparation (Supplementary Fig. 25c, 25d)[67]. These data suggested that SEM is not appropriate to evaluate the in situ structure of the composite hydrogel.

The bCAII-responsiveness of the **APmoc-F(CF₃)F** fibers retained intact in the composite hydrogel as confirmed by rheological and HPLC analysis. The rheological measurements indicated that the storage modulus ($G'$) decreased to 770 Pa upon addition of bCAII, while the value for $G'$ did not change (2300 Pa) on addition of buffer lacking bCAII (Fig. 3d, Supplementary Fig. 27). Importantly, bCAII treatment changed the tan δ value of the composite gel from 0.096 to 0.073, which was similar to the value for the single-component agarose gel (0.080) (Supplementary Figs. 20d, 27e). HPLC analysis also indicated that 76% of **APmoc-F(CF₃)F** was decomposed (Supplementary Fig. 28). Furthermore, we embedded the bCAII/**EAT(avidin)** set in the composite hydrogel and confirmed that degradation of the **APmoc-F(CF₃)F** nanofibers occurred in response to avidin using HPLC, that is 45% of **APmoc-F(CF₃)F** was decomposed in the composite, which was almost the same amount as in the single-component **APmoc-F(CF₃)F** gel (40%), indicating that the protein-triggered bCAII activation can work well even in the presence of agarose (Supplementary Fig. 29).

**Controlled protein release in response to a target protein from the composite hydrogel.** We subsequently investigated protein entrapment and controlled release according to the method shown in Fig. 5a. The protein entrapment capability of the composite hydrogel was evaluated using myoglobin (Mb) as a model protein. The composite hydrogel (Mb/bCAII/**EAT(avidin)**⊂**APmoc-F(CF₃)F**/agarose) was prepared by addition of a Mb solution into a mixture of agarose (0.5 wt%) and **APmoc-F(CF₃)F** (0.6 wt%) before gelation. The resultant hydrogels were placed in vials after gelation, and buffer solution was carefully added on top of the hydrogels, followed by SDS-PAGE analyses of the supernatants 3 h after incubation. As shown in Fig. 5b and Supplementary Fig. 30, the entrapment ratio greatly improved with **APmoc-F(CF₃)F** in the composite hydrogels (97%), while the agarose hydrogel alone exhibited poor entrapment capability (6.6%). This result suggested that the **APmoc-F(CF₃)F** nanofiber networks were the main contributor to the Mb entrapment. The single-component **APmoc-F(CF₃)F** hydrogel could not be taken out of the mold due to its poor mechanical properties. CLSM imaging of the composite hydrogel using Alexa fluor 647-modified Mb (Ax647-Mb) clearly visualized the fibrous morphology of the hydrogel, which was well overlapped with **APmoc-F(CF₃)F** fibers stained with **TMR-Gua** (Pearson's correlation coefficient: 0.58) (Fig. 5c). In contrast, the Ax647-Mb image was not colocalized with that of FL-agarose. Finally, we conducted protein-responsive protein release experiments using the composite hydrogel system (Mb/bCAII/**EAT(avidin)**⊂**APmoc-F(CF₃)F**/agarose). The avidin solution was added to the composite hydrogel, incubated, and the supernatants were analyzed by SDS-PAGE (Fig. 5a). The results showed that 75% of the Mb was released from the hydrogel, whereas a lower amount of Mb was

released under the buffer treatment (2.3%, Fig. 5d, Supplementary Fig. 34). Thus, the release of Mb from the composite hydrogel was modulated by avidin. The folding structure of the released Mb retained unchanged as confirmed by UV–vis absorption spectroscopy (Supplementary Fig. 35). We also succeeded in the release rate of RNase A, a protein-based drug candidate for cancer therapy[68], could be enhanced in response to the presence of anti-DNP IgG or avidin (Fig. 5e, Supplementary Figs. 36, 37, 38). We confirmed that the released RNase A was active (Supplementary Fig. 39). Taken together, it is demonstrated that our composite system (bCAII/EAT⊂**APmoc-F(CF₃)F**/agarose) can function as a non-enzymatic protein-responsive protein release matrix, which can potentially be used for the release of protein-based pharmaceuticals controlled by a distinct biomarker protein.

## Discussion

To date, structural analysis of the composite hydrogels has been mainly conducted by SEM, which may concern about artifacts derived from the gel drying process[67] and has a poor capability to distinguish between the similar morphologies of supramolecular nanofibers and polymer networks. In situ CLSM imaging enables the visualization of an orthogonal network of supramolecular nanofibers and agarose with distinct morphologies in the gel state. Furthermore, we successfully evaluated the in-depth relationship between the network structure and its functions. Our hydrogels acted as protein-responsive protein release matrices, which highlights the potential to program sophisticated functions into a composite hydrogel. Such composite hydrogels should be promising as a unique matrix embedded in a body, which is capable of autonomously releasing protein biopharmaceuticals in response to increase of non-enzymatic biomarker proteins. Our concept that integrates distinct chemical systems in a well-controlled manner provides information toward producing next-generation intelligent soft materials for drug delivery, cell culturing, and regenerative medicine.

## Methods

**bCAII response on the hydrogel array chip.** A suspension of an **APmoc-F(CF₃)F** or **Bz-FF** powder in 100 mM HEPES, pH 8.0 was heated by a heating gun (PJ-206A1, Ishizaki) until dissolving. After cooling to rt, 20 µL of the resultant solution was added to a glass plate and incubated for 2 h at 25 °C in a humid container to avoid dryness. 6 h after addition of bCAII solution (Sigma-Aldrich, C2522, 100 µM, 2.0 µL) with/without EZA (0 or 1.0 mM), the samples were touched with a paper (prowipe S220, elleair) to judge whether the samples were gel or sol. Paper absorb sol samples, but not gel samples. The assay conditions were referred in the figure captions.

**Protein response on the hydrogel array chip.** A suspension of an **APmoc-F(CF₃)F** powder in 100 mM HEPES, pH 8.0 was heated by a heating gun until dissolving. After cooling to rt, 18.0 µL of **APmoc-F(CF₃)F** solution was added to a glass plate. A couple of minutes after addition on a glass chip (sample temperature was ca. 30 °C), the mixture of bCAII and EAT (2.0 µL) was added before gelation. After 2 h, protein solutions (avidin (neutralized): Wako Pure Chemical, 015-24231, 200 µM, 2.0 µL; DHFR: 450 µM, 2.0 µL; anti-DNP IgG: Thermo Fisher Scientific, 04-8300, 150 µM, 2.0 µL) were added. After incubation for 6, 12, or 18 h at 25 °C in a humid container to avoid dryness, the samples were touched with a paper (prowipe S220, elleair) to judge whether the samples were gel or sol. Paper absorb sol samples, but not gel samples. The assay conditions were referred in the figure captions.

**HPLC analysis of the hydrogel droplets.** Six hour after addition of bCAII or avidin, the samples were diluted by a mixture of 1:1 CH₃CN/H₂O (60 µL) and a DMSO solution of terephthalic acid (100 mM, 2 µL). The resultant mixture was filtered with membrane filter (diameter: 0.45 µm), and then analyzed by RP-HPLC (column: YMC-Triart C18, A:B = 10:90 to 80:20 for 40 min, A: CH₃CN containing 0.1% TFA, B: H₂O containing 0.1% TFA).

**CLSM imaging of the APmoc-F(CF₃)F hydrogel.** The suspension of **APmoc-F(CF₃)F** in 100 mM HEPES, pH 8.0 (0.35 wt%) was heated by a heating gun until dissolving. After cooling to rt, the resultant mixture (18 µL) was transferred to a glass bottom dish (Matsunami). A couple of minutes after addition on a glass chip (sample temperature was ca. 30 °C), the mixture of bCAII and **EAT(avidin)** (100 µM, 200 µM, respectively, 2 µL in 100 mM HEPES, pH 8.0) was added and incubated for 15 min at

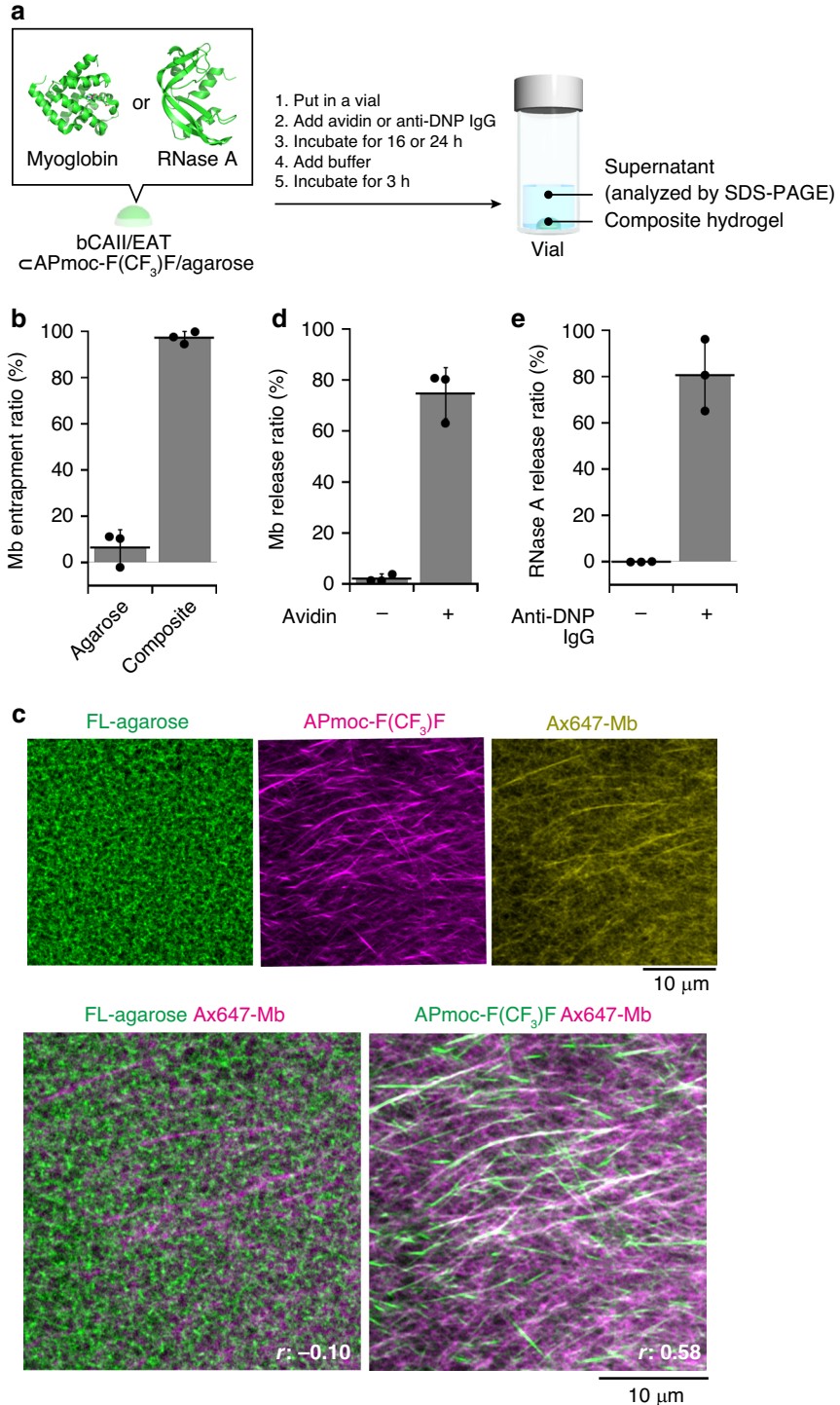

**Fig. 5 Protein-responsive protein release from the composite hydrogel. a** Schematic illustration of the protocol of protein-responsive protein release from the composite hydrogel. PDB ID: 5ZZE (myoglobin, Mb) and 1AFK (RNase A). **b** The Mb entrapment ratio of agarose and composite hydrogels ($n = 3$). **APmoc-F(CF₃)F** hydrogel could not be taken out of the mold. **c** High-resolution Airyscan CLSM images of the composite hydrogel containing Ax647-Mb (Ax647-Mb⊂**APmoc-F(CF₃)F**/FL-agarose/**TMR-Gua**). *r*: Pearson's correlation coefficient. As shown in Supplementary Fig. 31, Ax647-Mb was trapped more efficiently in the composite hydrogel than the agarose hydrogel. The fibrous morphology of the Alexa Fluor 647 dye could not been observed, implying that interaction between **APmoc-F(CF₃)F** fibers and Mb is crucial (Supplementary Fig. 32). Please see Supplementary Fig. 33 for other control conditions. **d** Avidin-responsive Mb release and (**e**) anti-DNP IgG-responsive RNase A release from the composite hydrogel ($n = 3$). Condition: [**APmoc-F(CF₃)F**] = 0.6 wt%, [agarose] = 0.5 wt% (for **b**, **d** and **e**), [FL-agarose] = 0.5 wt% (for **c**), [Mb] = 36 μM (for **b** and **d**), [Ax647-Mb] = 36 μM (for **c**), [RNase A] = 0.25 mg/mL (for **e**), [**TMR-Gua**] = 10 μM (for **c**), [bCAII] = 10 μM, [**EAT(avidin)**] = 20 μM (for **d**), [avidin] = 20 μM (for **d**), [**EAT(DNP IgG)**] = 15 μM (for **e**), [anti-DNP IgG] = 15 μM (for **e**), 100 mM HEPES, pH 8.0. The data represent the mean ± standard deviation.

rt. To the resultant hydrogel, a solution of avidin (200 μM, 2 μL), buffer, or avidin premixed with biotin (200 μM, 1.2 mM, respectively, 2 μL) was added. After incubation for 6 h, a solution of **TMR-Gua** (280 μM, 1 μL, 1:9 DMSO/100 mM HEPES (pH 8.0)) was added, and CLSM imaging was subsequently conducted.

**Preparation of the composite hydrogel containing bCAII/EAT**. A suspension of an agarose powder in 100 mM HEPES, pH 8.0 was heated for 5 min by heating until dissolving. The hot agarose solution was added to an **APmoc-F(CF₃)F** powder, and the mixture was heated by a heating gun until dissolving. The resultant mixture (200 μL) was transferred to a vial or a PDMS mold (pore diameter: 10 mm). Before gelation, a mixture of bCAII and EAT (20 μL) was added. After incubation for 2 h in a humid container to avoid dryness, the resultant composite hydrogel was used for response tests and rheological experiments.

**CLSM imaging of the composite hydrogel**. A suspension of a FL-agarose powder in 100 mM HEPES, pH 8.0 was heated for 5 min by a heating gun until dissolving. The hot FL-agarose solution was added to an **APmoc-F(CF₃)F** powder. The resultant mixture was heated again by a heating gun until dissolving. Before gelation, the mixture (20 μL) was transferred to a glass bottom dish (Matsunami). After incubation for 15 min at rt, a solution of **TMR-Gua** (280 μM, 1 μL, 1:9 DMSO/100 mM HEPES (pH 8.0)) was added to the composite hydrogel, and then CLSM imaging was carried out. The detailed assay conditions were referred in the figure captions.

**HPLC analysis of the composite hydrogel**. The stock solution of avidin (200 μM, 20 μL) was added to the composite hydrogel. After incubation at rt for 16 h, DMF (600 μL) and a DMSO solution of terephthalic acid (100 mM, 30 μL) were added to the composite hydrogel, and the resultant mixture was dissolved by vortex mixing. The mixture was filtered and then analyzed by RP-HPLC (column: YMC-Triart C18, A:B = 10:90 to 80:20 for 40 min, A: CH₃CN containing 0.1% TFA, B: H₂O containing 0.1% TFA).

**CD spectroscopy**. The sample was poured into a quartz cell before gelation (optical length: 0.05 mm). After incubation for 10 min at room temperature, CD spectra were measured.

**Scanning electron microscopy**. **APmoc-F(CF₃)F**, agarose, and composite hydrogels were frozen by immersing in liquid nitrogen and lyophilized overnight. The samples were put on a conductive carbon adhesive tape (thin aluminum foil core) and sputter-coated with a thin layer of platinum (ca. 5 nm). The secondary electron images were acquired by a field emission scanning electron microscope (Hitachi, SU8200) at a 1.5 kV voltage.

**Protein-release experiments**. Proteins were embedded in the composite hydrogel (20 μL) by addition of protein solutions before gelation (sample temperature was ca. 30 °C). After moving from the PDMS mold to a vial, the solution of avidin (200 μM, 2 μL) was added to the composite hydrogel. After incubation for 16 (for avidin) or 24 h (for RNase A) at rt in a humid container to avoid dryness, a HEPES buffer (100 mM, pH 8.0, 20 μL) was added to the resultant gel, and subsequently incubated for 3 h at rt. Ten microliters of supernatant was picked out, and mixed with a Laemmli buffer (5-times higher concentration containing 10 vol% 2-mercaptoethanol and 3 mM biotin). The resultant mixture was heated at 95 °C for 5 min, analyzed by SDS-PAGE, and quantified by ChemiDoc-XRS (observed at 595 nm). The unprocessed gel data were shown in the Source Data file.

**Rheological analysis**. The resultant disk-shaped composite hydrogels (ca. 10 mm) were carefully took out from the PDMS mold and put onto the stage of a rheometer (MCR-502, Anton Paar) with a parallel plate geometry. Strain-sweep data were obtained using shear mode at a frequency of 10 rad/s, and linear dynamic viscoelasticity were measured in shear mode at 1% strain amplitude for frequency sweep.

**UV–Vis absorption spectroscopy of FL-agarose**. FL-agarose (0.50 mg) was suspended in 10 mM tetraborate buffer (500 μL). The suspension was heated until dissolving with a heating gun for 5 min (concentration of repeating units: 3.26 mM). The resultant solution was measured by a UV–Vis spectrometer to determine absorbance derived from fluorescein to be 0.0365 (0.487 μM, molar absorption coefficient: 75,000[69]), corresponding to 0.015 mol% relative to the repeating unit of agarose (supplementary Fig. 40).

**Modification of myoglobin with Alexa fluor 647**. To a PBS solution of myoglobin (Sigma-Aldrich, M1882, 0.5 mg/mL, 6 mL, pH 8.0) was added a DMSO solution of Alexa fluor 647-NHS ester (Thermo Fisher Scientific, A-20006, 50 mM, 6.72 μL). The reaction mixture was incubated at 4 °C for 24 h. The resulting mixture was diluted with PBS (18 mL) and dialyzed by Spectra/Por dialysis membrane (MWCO 8000) against PBS (500 mL, 3 times) and 100 mM HEPES (1 L, pH 8.0). The solution was concentrated by an Amicon-Ultra Centrifugal filter unit (NMWL 3500) to obtain Ax647-Mb as a blue transparent solution (Mb: 718 μM, Alexa fluor 647: 775 μM

determined by UV-vis absorption spectroscopy in supplementary Fig. 41). The molar absorption coefficients of Mb (18,800[70]) and Alexa Flour 647 (290,000[69]) were used.

**Determination of the activity of RNase A**. The enzymatic activity of RNase A was monitored by DNase+RNase detection kit (Jena Bioscience). The supernatant solution was diluted by 1000-fold with a detection buffer. The resultant solution (10 μL) and the master mix containing a probe (40 μL) were mixed on ice. The time course of fluorescent intensity was monitored by a plate reader (infinite M200, TECAN, excitation wavelength: 495 nm, emission wavelength: 520 nm, gain: 100, interval: 1 min, temperature: 37 °C).

## Data availability
The authors declare that the data supporting the findings of this study are available with the paper and its Supplementary information files. The data that support the findings of this study are available from the corresponding author upon reasonable request. Source data are provided with this paper.

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

## Acknowledgements

The authors thank T. Tamura (Kyoto University) and T. Yoshii (Nagoya Institute of Technology) for his assistance for synthesis of EAT(avidin, long) and TMR-Gua, respectively. We appreciate K. Okamoto-Furuta and H. Kohda (Kyoto University) for technical assistance in preparation of SEM samples. Victoria Muir, PhD, from Edanz Group (www.edanzediting.com/ac) edited a draft of this manuscript. This work was supported by a Grant-in-Aid for Scientific Research on Innovative Areas "Chemistry for Multimolecular Crowding Biosystems" (JSPS KAKENHI Grant JP17H06348), JST ERATO Grant Number JPMJER1802 to I.H., and by a Grant-in-Aid for Young Scientists (JSPS KAKENHI Grant JP18K14333 and JP20K15400) to R.K. H.S. acknowledge JSPS Research Fellowships for Young Scientists (JP16J10716).

## Author contributions

I.H. supervised the project. I.H. and H.S. designed the experiments of the protein-responsive supramolecular hydrogel. I.H. and R.K. designed the experiments of the supramolecular/agarose composite hydrogel. K.N. performed all experiments. T.M. found APmoc-F(CF$_3$)F as an enzyme-sensitive hydrogel. K.N., S.M., T.A., and K.U. performed rheological measurements. I.H. and R.K. wrote the manuscript, with helpful comments from all authors.

## Competing interests

The authors declare no competing interests.
