## [Peer Review File · Nature Communications]

Reviewers' Comments:

Reviewer #1:

Remarks to the Author:

The authors have revised the manuscript taking into account all of the previous reviewers' comments. I am satisfied by the changes made and support this paper being published as is.

Reviewer #2:

Remarks to the Author:

The manuscript describes a novel composite hydrogel capable of controlled release triggered with non-enzymatic proteins. This is done by introducing a protein binding enzyme inhibitor conjugated to a protein ligand for a target protein which upon binding triggers enzyme activity for a gel to sol transition. A high degree of specificity was demonstrated for the transition in a matter of hours using this particular strategy. The concept is interesting and the work described provides proof-of-principle. The authors have made a number of improvements based on the comments provided during the first round of review but the following minor additions/corrections will further improve the manuscript for publication.

Specific Comments

- 1) Figure 3c and d is this $n=1$? Did the authors repeat these experiments and if so please provide number of repeats (n) and error bars (SD) to demonstrate the reproducibility of the data given that the authors use up to 4 significant figures while describing rheology data (for example lines 79 pg 13).
- 2) Page 18, line 22. Please correct text to ... "could not be taken out of the mold due to its poor mechanical properties."
- 3) Page 22, line 4. "To the best of our knowledge, this is the first example of...". We agree that CLSM imaging is an important technique in the structural studies of supramolecular composite hydrogels. However, this sentence seems distracting and it may not be closely related to the key novelty in this paper which is the design of the EAT molecule which endows the responsiveness of the supramolecular hydrogel to non-enzymatic proteins. It is suggested that this sentence should be removed or rephrased.

Reviewer #3:

Remarks to the Author:

The authors have addressed most of the issues raised by this reviewer as well as other reviewers in this submission. They have addressed the issue of novelty: the relationship between network structure and functionality + unique autonomous protein release. The authors also address the proof-of-concept use of these supramolecular hydrogels for the release of RNase A in response to anti-DNP IgG. All the control experiments and orthogonal assemblies, which were not described in detail previously, are now clearly explained in this manuscript. It is clear now that the protein structure, activity, and mechanical integrity in the gel is retained before and after orthogonal assembly. There are some amplification possibilities with this system and therefore this system is more sensitive than other protein responsive hydrogel and can be used even in low concentration of the stimulus protein.

I am happy with the method in which all the issues have been addressed and suggest accepting this manuscript.

REVIEWERS' COMMENTS:

Reviewer #1:

The authors have revised the manuscript taking into account all of the previous reviewers' comments. I am satisfied by the changes made and support this paper being published as is.

Reply:

We appreciate your reviewing and positive comment.

Reviewer #2:

The manuscript describes a novel composite hydrogel capable of controlled release triggered with non-enzymatic proteins. This is done by introducing a protein binding enzyme inhibitor conjugated to a protein ligand for a target protein which upon binding triggers enzyme activity for a gel to sol transition. A high degree of specificity was demonstrated for the transition in a matter of hours using this particular strategy. The concept is interesting and the work described provides proof-of-principle. The authors have made a number of improvements based on the comments provided during the first round of review but the following minor additions/corrections will further improve the manuscript for publication.

Reply:

We appreciate our positive comments. According to your concerns, we amended our manuscript as shown below.

Comment 1:

Figure 3c and d is this n=1? Did the authors repeat these experiments and if so please provide number of repeats (n) and error bars (SD) to demonstrate the reproducibility of the data given that the authors use up to 4 significant figures while describing rheology data (for example lines 7-9 pg 13).

Reply:

We newly conducted rheological measurement. We modified Fig. 3c, 3d to show the individual data and amended the main text as shown below.

Modification in the main text

Page 9, line 7:

Rheological analyses quantitatively demonstrated that the storage modulus (G') of the composite hydrogel was synergistically enhanced to be 2000 Pa, which was higher than that of the agarose hydrogel (530 Pa) or supramolecular **APmoc-F(CF₃)F** hydrogel (400 Pa) (Fig. 3c, 3e, Supplementary Fig. 20).

Page 11, line 10:

The rheological measurements indicated that the storage modulus (G') decreased to 770 Pa upon addition of bCAII, while the value for G' did not change (2300 Pa) on addition of buffer lacking bCAII (Fig. 3d, Supplementary Fig. 27).

Comment 2:

Page 18, line 22. Please correct text to ... “could not be taken out of the mold due to its poor mechanical properties.”

Reply:

According to your suggestion, we modified the main text as shown below.

Modifications in the main text

Page 12, line 10:

The single component APmoc-F(CF₃)F hydrogel **could not be taken out of the mold due to its poor mechanical properties.**

Comment 3:

Page 22, line 4. “To the best of our knowledge, this is the first example of...”. We agree that CLSM imaging is an important technique in the structural studies of supramolecular composite hydrogels. However, this sentence seems distracting and it may not be closely related to the key novelty in this paper which is the design of the EAT molecule which endows the responsiveness of the supramolecular hydrogel to non-enzymatic proteins. It is suggested that this sentence should be removed or rephrased.

Reply:

According to your comment, we modified the sentence as shown below.

Modifications in the main text

Page 13, line 14:

In situ CLSM imaging enables the visualization of an orthogonal network of supramolecular nanofibers and agarose with distinct morphologies in the gel state.

Reviewer #3:

The authors have addressed most of the issues raised by this reviewer as well as other reviewers in this submission. They have addressed the issue of novelty: the relationship between network structure and functionality + unique autonomous protein release. The authors also address the proof-of-concept use of these supramolecular hydrogels for the release of RNase A in response to anti-DNP IgG. All the control experiments and orthogonal assemblies, which were not described in detail previously, are now clearly explained in this manuscript. It clear now that the protein structure, activity, and mechanical integrity in the gel is retained before and after orthogonal assembly. There is some amplification possibilities with this system and therefore this system is more sensitive than other protein responsive hydrogel and can be used even in low concentration of the stimulus protein.

I am happy with the method in which all the issues have been addressed and suggest accepting this manuscript.

Reply:

We appreciate your reviewing and positive comment.